# Lagrangians of Multiannual Growth Systems

**Petri P. Kärenlampi** 

Lehtoi Research, 81235 Lehtoi, Finland; petri.karenlampi@professori.fi

**Abstract:** Multiannual growth systems are modeled in generic terms and investigated using partial derivatives and Lagrange multipliers. Grown stock density and temperature sum are used as independent variables. Estate capitalization increases continuously with grown stock and temperature sum, whereas capital return rate and gross profit rate reach a maximum with respect to grown stock. As two restrictions are applied simultaneously, the results mostly but not always follow intuition. The derivative of capital return rate with respect to gross profit rate is positive, and negative with respect to capitalization. The derivative of capitalization with respect to capital return rate shows some positive values, as well as that with respect to gross profit rate. The derivative of the gross profit rate is positive with respect to both capitalization and capital return rate. The results indicate a variety of alternative strategies, which may or may not be multiobjective.

**Keywords:** gross profit rate; capitalization; capital return rate; expected value; probability density

## 1. Introduction

Growing biological systems provide ecosystem services, corresponding to crops, air purification, carbon storage, biodiversity, and protection from hazards such as erosion and avalanches [1,2]. Due to the variety of necessary ecosystem services, multiobjective management has gained popularity [3–9]. Our primary focus here is on forestry, but we believe the results can be applied or adapted to estates growing other multiannual plants such as bamboo or fruit trees [10,11].

The principal microeconomic objective in capital investments is the rate of return on capital [12–14]. Plant-growing estates can be seen as investments, possibly yielding a return on capital. Such return, however, can be partially operative and partially nonoperative. Operative return is based on owner activities, such as timber sales, whereas nonoperative occurs independent of owner activities, for example, as eventually increasing estate market values.

As the return on capital may be due to operations or capital valuations, there correspondingly are different possible strategies available. Straightforwardly, the operative capital return rate can be taken as an objective. On the other hand, value increment in the real estate market can be aspired [15,16]. In the latter case, one can simply select increasing estate value as the objective.

One can then ask, what is the role of profit in microeconomics. Profit naturally contributes to the operative capital return rate, but it hardly suffices as an independent measure of performance in capital-intensive businesses [12–14]. There are, however, circumstances where the gross profit rate is of particular interest [17,18], even if not necessarily microeconomic interest. A gross income stream is partially used to pay taxes, and the goods produced serve as input to the regional economy. In the case of forestry, the wood raw material produced enables industries, as well as services the industries need. In other words, the gross profit rate may suffice as an objective to public agents.

Lagrange multipliers have been previously used to optimize sampling in measurements of the timber stock [19]. Eulerian–Lagrangian extremization has been used in the gasification of forest residues [20]. Lagrange multipliers have been used in a relaxation approach regarding road network placement and harvesting machinery location [21].

In this paper, we intend to present a simple, generic model of estates growing multiannual plants using hyperbolic growth functions and Lagrange multipliers. Our focus is on economical quantities. We first present a simple expression for the capitalization of forest estates, as well as for the gross profit rate and capital return rate. Grown stock density and temperature sum are taken as independent variables. The capital return rate is extremized, with gross profit rate and capitalization as restrictions. In the second stage, capitalization is taken as the main objective function, and capital return rate and gross profit rate as restrictions. Third, the gross profit rate is maximized, with capitalization and capital return rate as restrictions. Lagrange multipliers, determining the total derivative of any objective function with respect to any restriction, are determined. Generality is aspired—however, numerical solution with two restrictions requires fixing parameters to describe European boreal forestry.

## 2. Materials and Methods

Estates growing multiannual plants are here approached in terms of statics. In other words, an idealized steady-state system is discussed. Even if trees grow annually, a forest estate may remain in a stationary state provided that the amount of annual growth is harvested annually. Stationarity within an estate may require an even stand age distribution [22]. Such a requirement, however, vanishes if expected values of observables are discussed within a stand [23]. Average grown stocking, estate value per hectare, gross profit rate, and operative capital return rate within a rotation are to be discussed, as well as the corresponding average growing season temperature sum.

Let us first establish a measure for the independent variable of average grown stock $V$. It is measured in monetary terms, the average current value of the grown plants per area unit. Instead of being a random variable, the expected value of the grown stock is determined by agent activity. It necessarily fluctuates in time, since harvesting occurs discretely, instead of continuously. Such fluctuation, however, is considered unimportant in the present context.

The grown stock contributes to capitalization per unit area, given as

$$C = B + (1 + u)V \tag{1}$$

where $B$ corresponds to bare land value and $u$ to eventual market premium in the real estate market [15,16]. It is worth noting that all dependent variables are here discussed in terms of expected values and given in terms of deterministic equations. Only the expected values are of interest since fluctuations of the dependent variables tend to average out along with time.

The average gross profit rate is given in terms of the relative value growth rate of plants, multiplied by the average grown stock. For the former quantity, a simple monotonically decreasing function is adopted as $a[1 - tanh(kV)]$, where $a$ and $k$ are scale factors. Correspondingly, the average gross profit rate is modeled as

$$P = a[1 - tanh(kV)]V \tag{2}$$

Finally, the average operative capital return rate is given as

$$R = \frac{P}{C} \tag{3}$$

The capital return rate is denoted as operative since eventual estate value change, measurable by the market premium factor $u$, is assumed not to change, and neither is the bare land value $B$ affected by the grown stock $V$.

Let us then write the derivatives of the capitalization, the gross profit rate, and the operative capital return rate with respect to the grown stocking as

$$\frac{\partial C}{\partial V} = 1 + u \tag{4}$$

$$\frac{\partial P}{\partial V} = a\left\{[1 - tanh(kV)] - kV\left[1 - tanh^2(kV)\right]\right\} \tag{5}$$

and

$$\frac{\partial R}{\partial V} = \frac{a\left\{B[1 - tanh(kV)] - kV\left[1 - tanh^2(kV)\right][B + (1+u)V]\right\}}{[B + (1+u)V]^2} \tag{6}$$

Above, the grown stock $V$ is used as the independent variable. Such a representation may be an oversimplification. The average growing season temperature sum is now taken as another independent variable. The temperature sum is predominantly determined by estate location, but it may vary with time. Correspondingly, any agent is able to contribute to this variable by estate acquisition. The temperature sum is introduced into the scale factor $a$ in the formula for the gross profit rate as

$$a = \alpha tanh\left(\frac{S - \varsigma}{\varphi}\right) \tag{7}$$

where $S$ is the average temperature sum, $\varsigma$ is a threshold value, and $\alpha$ and $\varphi$ are scale factors.

Once temperature sum is considered, it is plausible to assume that the bare land value depends on the temperature sum, most simply as

$$B = \beta S \tag{8}$$

where $S$ is the average temperature sum, and $\beta$ is a scale factor.

Let us then write the derivatives of the capitalization, the gross profit rate, and the operative capital return rate with respect to the temperature sum as

$$\frac{\partial C}{\partial S} = \beta \tag{9}$$

$$\frac{\partial P}{\partial S} = \alpha V[1 - tanh(kV)]\frac{1}{\varphi}\left[1 - \tanh^2\left(\frac{S - \varsigma}{\varphi}\right)\right] \tag{10}$$

and

$$\frac{\partial R}{\partial S} = \alpha V[1 - tanh(kV)]\left\{\frac{\frac{1}{\varphi}\left[1 - \tanh^2\left(\frac{S-\varsigma}{\varphi}\right)\right]}{\beta S + (1+u)V} - \frac{\beta\tanh\left(\frac{S-\varsigma}{\varphi}\right)}{[\beta S + (1+u)V]^2}\right\} \tag{11}$$

It is found from Equations (4) and (9) that the derivative of capitalization with respect to both grown stock and temperature sum is positive. The same applies to the derivative of gross profit rate with respect to temperature sum (Equation (10)), but the derivative with respect to grown stock changes from positive to negative with increasing grown stock (Equation (5)). The derivative of the operative capital return rate with respect to grown stock shows the same behavior, provided that the bare land value is nonzero (Equation (6)). The derivative of the operative capital return rate with respect to the temperature sum, in general, is positive but may reach negative values in some peculiar circumstances (Equation (11)).

The operative capital return rate as an objective function, with respect to a restriction in the gross profit rate and total estate value, results as a Lagrangian

$$\mathcal{L}_1 = R - \lambda_1(P - p) - \lambda_2(C - c) \tag{12}$$

where $p$ is the restriction for the gross profit rate $P$, and $c$ is the restriction for capitalization $C$. Extremization of Equation (12) in terms of $\frac{\partial \mathcal{L}_1}{\partial V} = \frac{\partial \mathcal{L}_1}{\partial S} = \frac{\partial \mathcal{L}_1}{\partial \lambda_1} = \frac{\partial \mathcal{L}_1}{\partial \lambda_2} = 0$ results as an

extremized Lagrangian $\mathcal{L}_1^*$. Then, $\lambda_1 = \frac{d\mathcal{L}_1^*}{dp} = \frac{dR^*}{dp} = \frac{dR^*}{dP^*}$, and $\lambda_2 = \frac{d\mathcal{L}_1^*}{dc} = \frac{dR^*}{dc} = \frac{dR^*}{dC^*}$. Here, all quantities with an asterisk (*) in the superscript refer to extremized quantities. Further, extremizing Lagrangians with respect to any Lagrange multiplier results as the extremized restriction function gaining the restriction value.

The estate capitalization as an objective function, with respect to a restriction in the capital return rate and the gross profit rate, results as a Lagrangian

$$\mathcal{L}_2 = C - \lambda_3(R - r) - \lambda_4(P - p) \tag{13}$$

where $r$ is the restriction for the operative capital return rate $R$. Extremization of Equation (13) results as an extremized Lagrangian $\mathcal{L}_2^*$. Then, $\lambda_3 = \frac{d\mathcal{L}_2^*}{dr} = \frac{dC^*}{dr} = \frac{dC^*}{dR^*}$, and $\lambda_4 = \frac{d\mathcal{L}_1^*}{dp} = \frac{dC^*}{dp} = \frac{dC^*}{dP^*}$.

Finally, the gross profit rate as an objective function, with respect to a restriction in the capitalization and the capital return rate, results as a Lagrangian

$$\mathcal{L}_3 = P - \lambda_5(C - c) - \lambda_6(R - r) \tag{14}$$

where $r$ is the restriction for the operative capital return rate $R$. Extremization of Equation (14) results as an extremized Lagrangian. Then, $\lambda_5 = \frac{d\mathcal{L}_3^*}{dc} = \frac{dP^*}{dc} = \frac{dP^*}{dC^*}$, and $\lambda_6 = \frac{d\mathcal{L}_3^*}{dr} = \frac{dP^*}{dr} = \frac{dP^*}{dR^*}$.

For all the expressions above, analytical solutions exist. However, many of them would not be easy to interpret. Numerical solutions, instead, can be graphically illustrated. Numerical solutions require parameter values—fixing parameter values, however, reduce the generality of the treatment. Here, reduced generality is accepted, and a set of parameter values, corresponding to European boreal conditions, is adopted as given in Table 1. The parametrization in Table 1 is designed simply by matching Equations (2) and (8) with boreal growth and yield models [24–29], as well as financial data from the region [30,31].

**Table 1.** Parameter values used in numerical solutions.

| Symbol | Value | Unit | Name |
|--------|-------|------|------|
| $u$ | 0.3 | | Relative estate market premium |
| $k$ | 0.000101 | | Grown stocking scale factor |
| $\alpha$ | 0.1 | | Relative growth scale factor |
| $\varsigma$ | 600 | Degrees C | Thermal sum threshold value |
| $\varphi$ | 450 | Degrees C | Thermal sum scale factor |
| $\beta$ | 0.7 | Eur/(ha*degrees C) | Bare land value scale factor |

## 3. Results

First, the estate capitalization, given in Equation (1), consisting of two terms, is dominated by the latter term, given the parameter values in Table 1. The estate capitalization in relation to the grown stock value in turn is dominated by the parameter $u$, corresponding to estate market premium, in relation to grown stock value.

The expected value of gross profit rate, as given in Equations (2) and (7), is shown in Figure 1. The expected value of operative capital return rate, as given in Equation (3), is shown in Figure 2. It is found that both the gross profit rate and the capital return rate depend strongly on the temperature sum. Both reach a maximum as a function of grown stock, but the capital return rate reaches the maximum at a much lower stock than the gross profit rate (Figures 1 and 2).

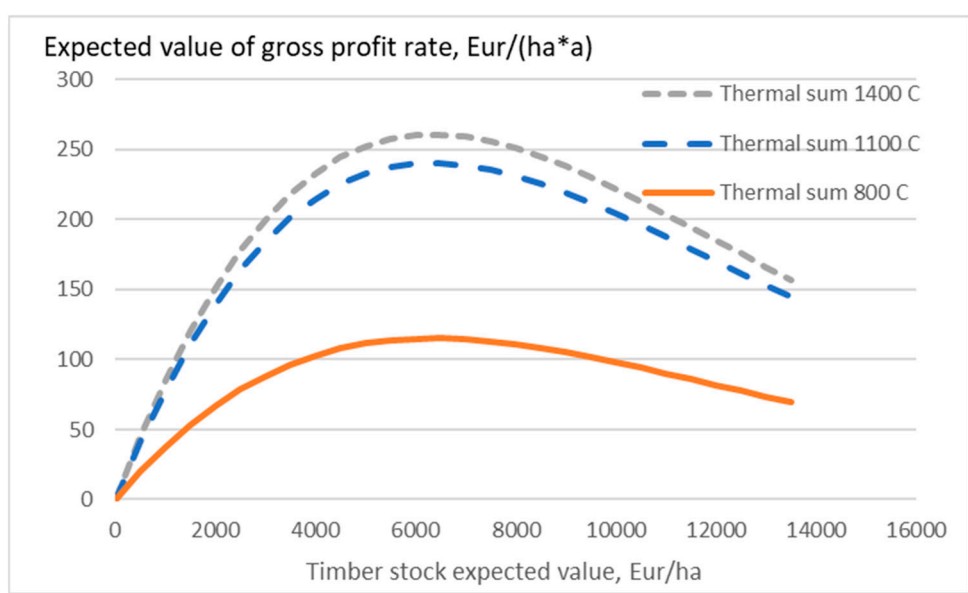

**Figure 1.** Expected value of gross profit rate as a function of grown stock and temperature sum.

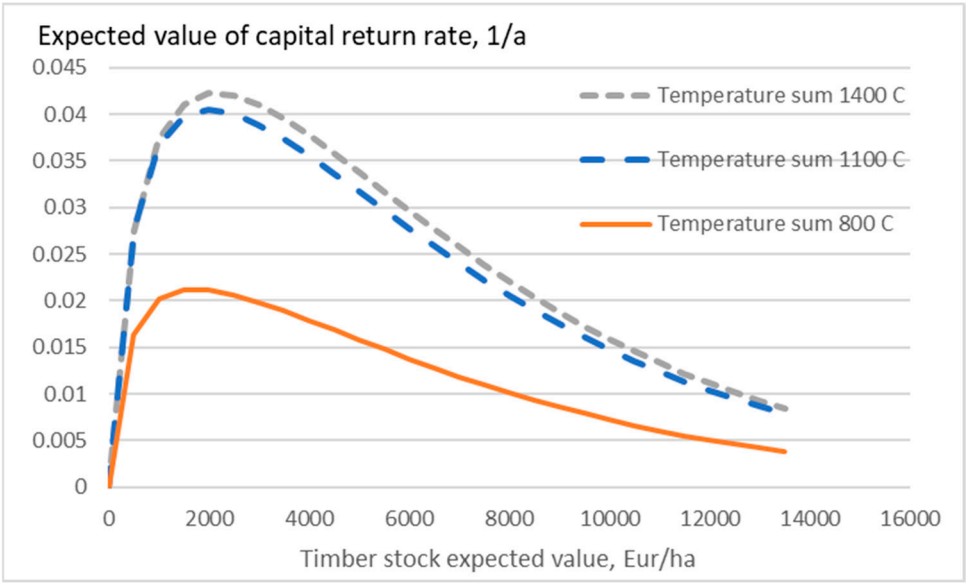

**Figure 2.** Expected value of operative capital return rate as a function of grown stock and temperature sum.

Figures 1 and 2 also readily reveal that the derivative of the gross profit rate and the operative capital return rate are both positive at low grown stock and negative at large grown stock, as indicated in Equations (5) and (6). The sign change happens at much lower grown stock in the case of the operative capital return rate. Equation (4) indicates that the derivative of the capitalization with respect to grown stock is constant, corresponding to $(1 + u)$.

Figures 3 and 4 show the derivatives of gross profit rate and operative capital return rate, respectively, with respect to the temperature sum. The effect of the temperature sum is the greatest with low temperature sum. The derivative reaches the highest values in the range of grown stock where the affected property is of the highest value. The derivative of the gross profit rate is always positive, whereas the derivative of the capital return rate reaches also some negative values. The latter appear in the case of high temperature sum and low grown stock, where the grown stock does not dominate the denominators in Equation (11).

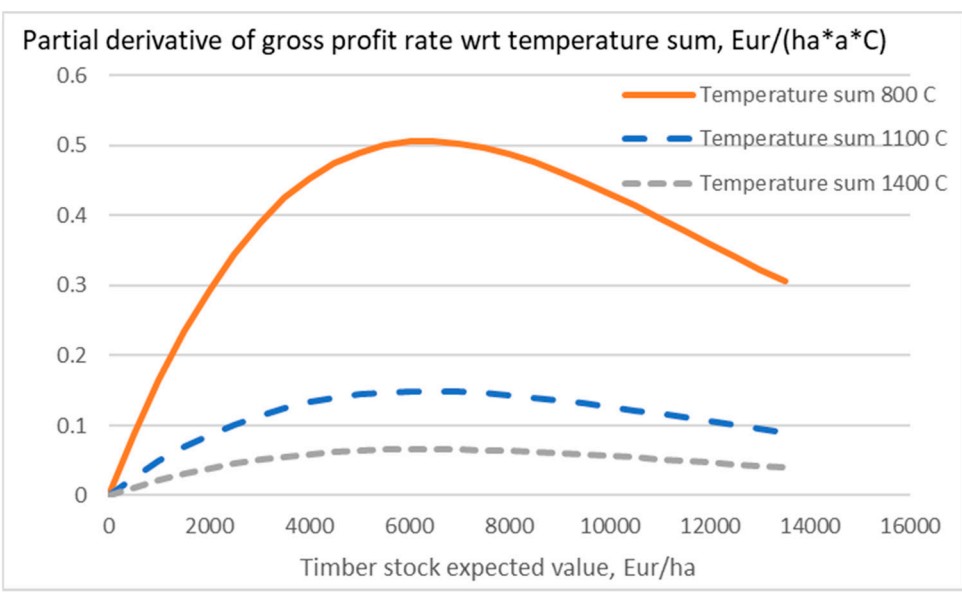

**Figure 3.** Partial derivative of gross profit rate with respect to temperature sum, according to Equation (10).

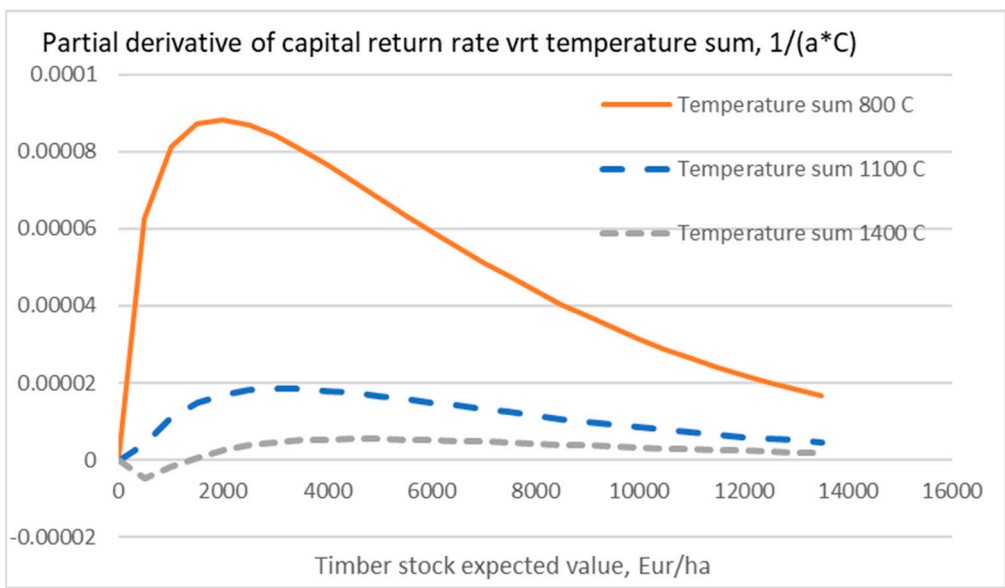

**Figure 4.** Partial derivative of operative capital return rate with respect to temperature sum, according to Equation (11).

Extremizing the Lagrangian of Equation (12) results as solutions of the Lagrange multipliers $\lambda_1 = \frac{dR^*}{dP^*}$ and $\lambda_2 = \frac{dR^*}{dC^*}$, corresponding to the total derivatives of the operative capital return rate under restrictions in both gross profit rate and capitalization. It is found from Figure 5 that the derivative with respect to the gross profit rate is always positive, and that with respect to capitalization is negative (Figure 6). The grown stock appearing in both the numerator and denominator of Equation (3), the magnitude of both of the derivatives vanish along with increasing grown stock.

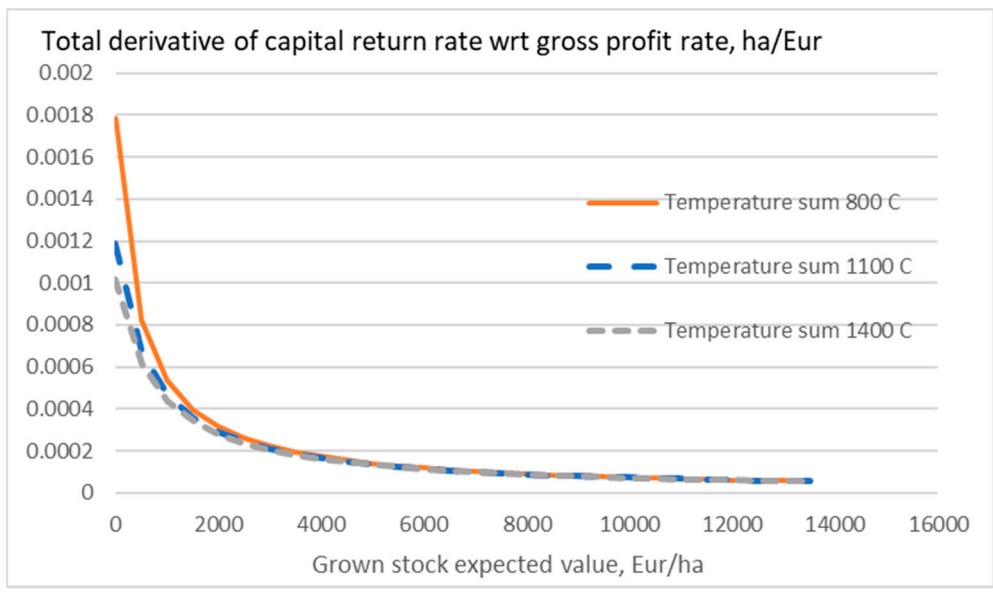

**Figure 5.** Total derivative of the operative capital return rate with respect to the gross profit rate, according to the Lagrangian of Equation (12) extremized.

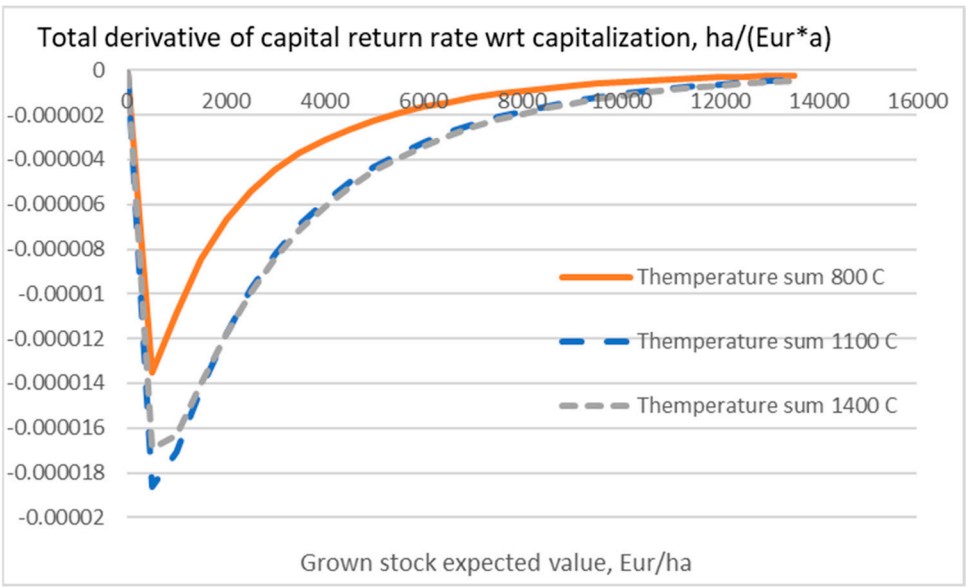

**Figure 6.** Total derivative of the operative capital return rate with respect to the capitalization, according to the Lagrangian of Equation (12) extremized.

Extremizing the Lagrangian of Equation (13) results as solutions of the Lagrange multipliers $\lambda_3 = \frac{dC^*}{dR^*}$ and $\lambda_4 = \frac{dC^*}{dP^*}$, corresponding to the total derivatives of the capitalization under restrictions in both capital return rate and gross profit rate. The derivative of the capitalization with respect to the operative capital return rate shows behavior somewhat less compliant with intuition: positive derivative values appear in Figure 7. This may be related to the fact that the partial derivatives of the capitalization and the operative capital return rate with respect to the grown stocking are of the same sign, unless the grown stocking is large (Equations (4) and (6)). The derivative of the capitalization with respect to the gross profit rate changes from positive to negative as the gross profit rate starts do diminish as a function of grown stocking (Figure 8).

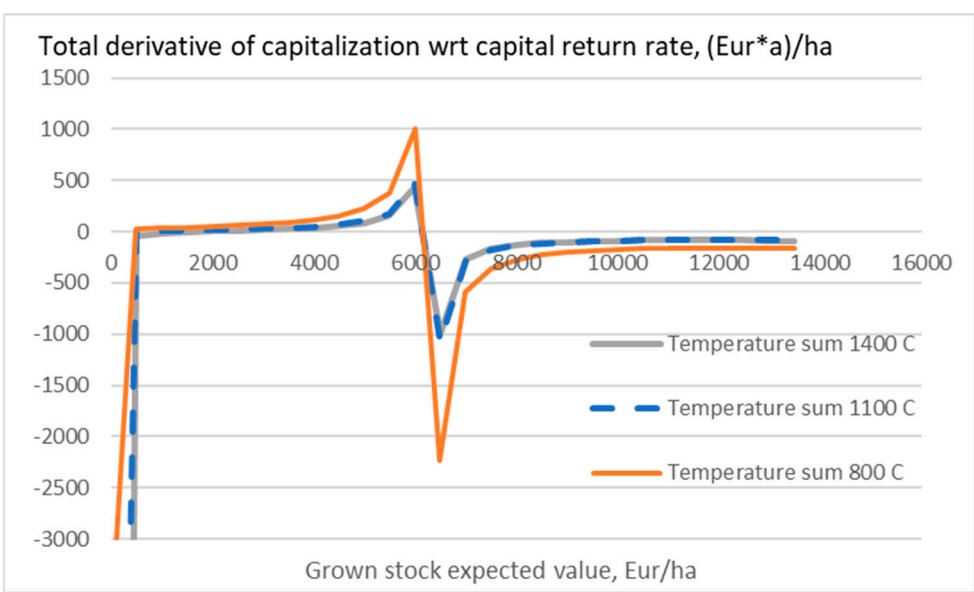

**Figure 7.** Total derivative of the capitalization with respect to the operative capital return rate, according to the Lagrangian of Equation (13) extremized.

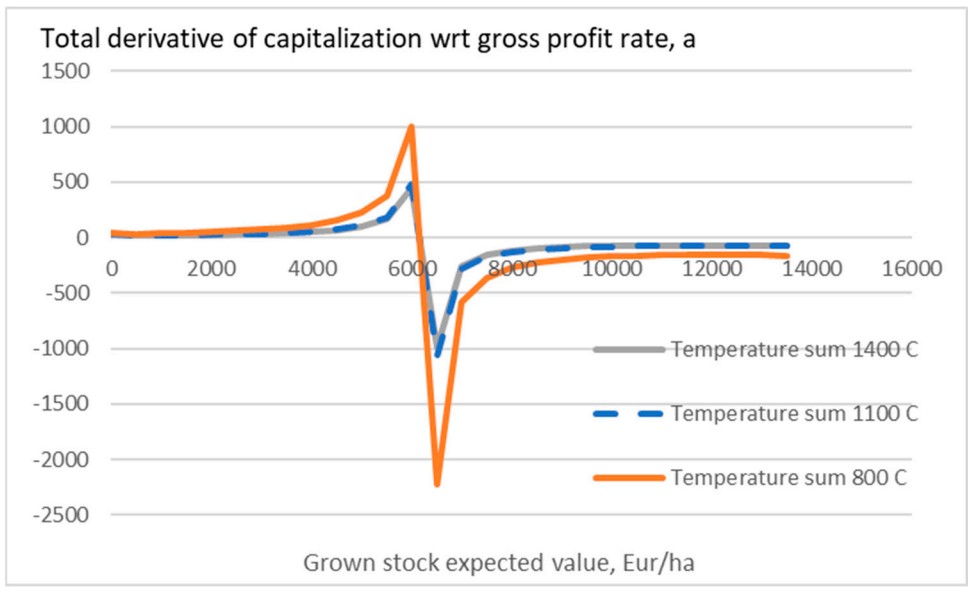

**Figure 8.** Total derivative of the capitalization with respect to the gross profit rate, according to the Lagrangian of Equation (13) extremized.

Extremizing the Lagrangian of Equation (14) results as solutions of the Lagrange multipliers $\lambda_5 = \frac{dP^*}{dC^*}$ and $\lambda_6 = \frac{dP^*}{dR^*}$, corresponding to the total derivatives of the gross profit rate under restrictions in both capitalization and capital return rate. Both of the total derivatives in Figures 9 and 10 are positive, $\lambda_5$ reaching a maximum with grown stock, $\lambda_6$ increasing apparently linearly.

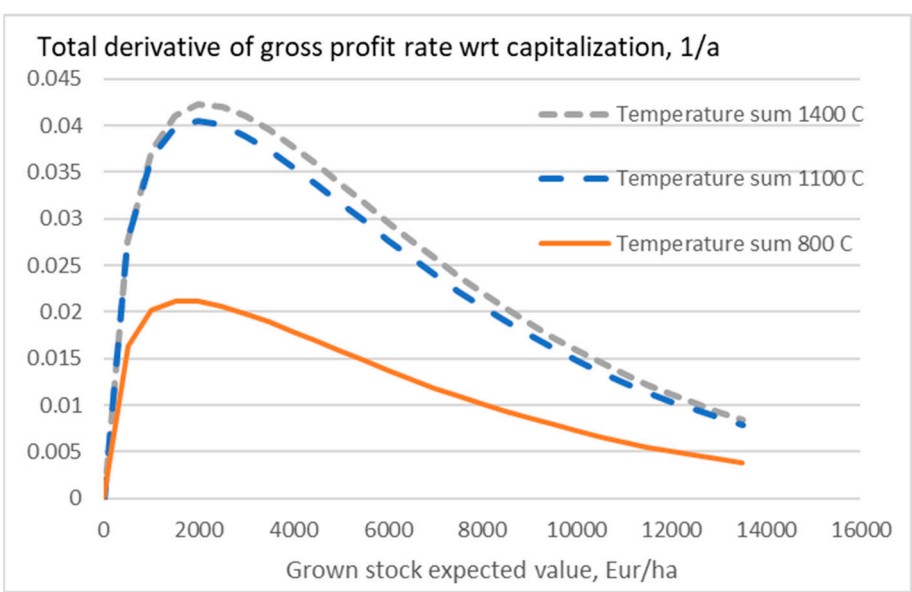

**Figure 9.** Total derivative of the gross profit rate with respect to capitalization, according to the Lagrangian of Equation (14) extremized.

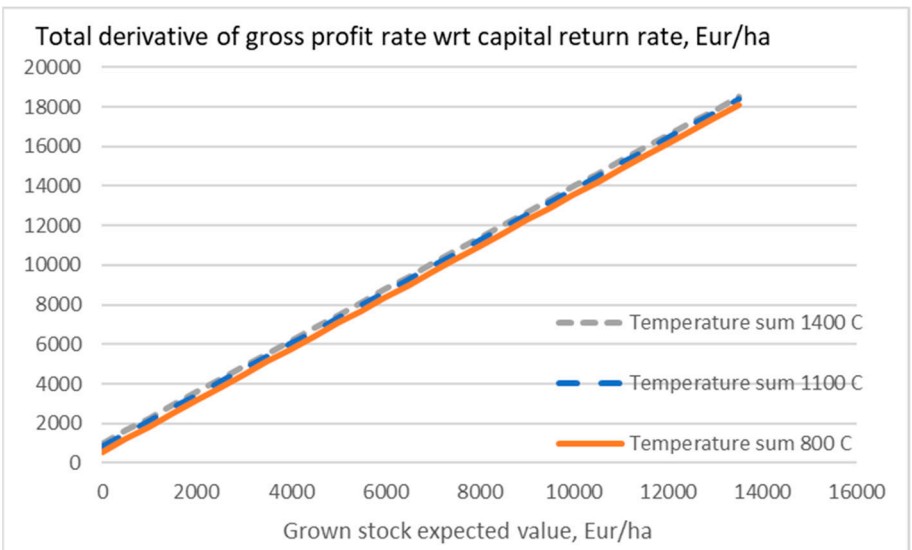

**Figure 10.** Total derivative of the gross profit rate with respect to the operative capital return rate, according to the Lagrangian of Equation (14) extremized.

## 4. Discussion

Three different objective functions have been discussed in this paper. The three different objective functions obviously correspond to three different strategies. The capital return rate appears the most natural objective function in capital economy [12–14]. A shortcoming is that the operative capital return rate only incorporates operative revenues and omits eventual development in estate valuation.

The second objective function, the estate capitalization, appears unconventional on first hearing. However, such an objective function may be essential if the intention is to create profits in the real estate market. The capitalization function appearing in Equation (1) has been proposed to apply in the Nordic countries and in North America [15,16,32,33], indicating that it may be profitable to acquire estates of low grown stock and divest estates of high grown stock.

The third objective function, the gross profit rate, appears the most peculiar. Profit contributes to the return rate of capital, but neglects capitalization, and thus hardly suffices

as an independent measure of financial performance. However, there are circumstances where the gross profit rate, reflecting the value stream of the production, may be of interest [17,18]. A gross income stream is partially used to pay taxes, and the goods produced serve as input to the regional economy. In the case of forestry, the wood raw material produced enables industries, as well as services the industries' need. In other words, the gross profit rate may suffice as an objective for public agents.

Considering any one of the three objective functions individually is not a very complicated task. Given objective functions in the form of Equations (1)–(3), their derivatives with respect to the grown stock and the temperature sum can be written as in Equations (4) to (11). Considering two objective functions simultaneously, their relationship can be clarified in terms of the chain rule of derivatives, using Equations (4) to (11).

Considering three objective functions simultaneously is a somewhat more complicated issue, here discussed in terms of Lagrange multipliers. Some of the results appear intuitive, whereas others are counterintuitive at first glance. Intuition can be supported by discussing partial derivatives between the objective functions, readily solvable from Equation (3).

The partial derivative of the capital return rate with respect to gross profit rate is always positive, and with respect to capitalization almost certainly negative (Equation (3)). The Lagrange multipliers $\lambda_1 = \frac{dR^*}{dP^*}$ and $\lambda_2 = \frac{dR^*}{dC^*}$, corresponding to the total derivatives of the operative capital return rate under restrictions in both gross profit rate and capitalization, display the same signs as the corresponding partial derivatives, in Figures 5 and 6.

The partial derivative of the capitalization with respect to the capital return rate is almost certainly negative and, with respect to the gross profit rate, is always positive (Equation (3)). The Lagrange multipliers $\lambda_3 = \frac{dC^*}{dR^*}$ and $\lambda_4 = \frac{dC^*}{dP^*}$, corresponding to the total derivatives of the capitalization under restrictions in both capital return rate and gross profit rate, however, both display positive and negative values in Figures 7 and 8. The appearance of positive values in Figure 7 is suspected to be related to the fact that the partial derivatives of the capitalization and the operative capital return rate with respect to the grown stock are of the same sign unless the grown stocking is large (Equations (4) and (6)). The appearance of negative values in Figure 8 is suspected to be related to the fact that the partial derivatives of the capitalization and the gross profit rate with respect to the grown stock are of different signs at large grown stock (Equations (4) and (5)).

The partial derivative of the gross profit rate with respect to capitalization corresponds to the capital return rate, according to Equation (3). The Lagrange multiplier $\lambda_5 = \frac{dP^*}{dC^*}$, corresponding to the total derivative of the gross profit rate under restrictions in both capitalization and capital return rate, produces results very much like the capital return rate. In other words, Figure 9 does not differ much from Figure 2. Similarly, the partial derivative of the gross profit rate with respect to capital return rate corresponds to the capitalization, according to Equation (3). The Lagrange multiplier $\lambda_6 = \frac{dP^*}{dR^*}$, corresponding to the total derivative of the gross profit rate under restrictions in both capitalization and capital return rate, does not differ much from the capitalization given in Equation (1).

One might ask whether Equation (3) would solve the entire problematics very straightforwardly. In the extremized state, the capital return rate indeed is

$$R^* = \frac{P^*}{C^*} = \frac{p}{c} \tag{15}$$

Then, the partial derivatives become straightforwardly

$$\frac{\partial R^*}{\partial P^*} = \frac{1}{C^*} = \frac{1}{c} \tag{16}$$

and

$$\frac{\partial R^*}{\partial C^*} = -\frac{R^*}{(C^*)^2} = -\frac{R^*}{c^2} \tag{17}$$

However, the Lagrange multipliers correspond to total derivatives, such as

$$\lambda_1 = \frac{dR^*}{dP^*} = \frac{dR^*}{dP^*}(V, S, p, c) \tag{18}$$

which can be expanded with the chain rule of partial derivatives

$$\lambda_1 = \frac{dR^*}{dP^*} = \frac{\partial R^*}{\partial V}\frac{\partial V}{\partial P^*} + \frac{\partial R^*}{\partial S}\frac{\partial S}{\partial P^*} + \frac{\partial R^*}{\partial p}\frac{\partial p}{\partial P^*} + \frac{\partial R^*}{\partial c}\frac{\partial c}{\partial P^*} \tag{19}$$

Then, one might ask whether the Lagrange multipliers could be determined directly using Equation (19). No, they cannot; the extremizing condition

$$\frac{\partial \mathcal{L}_1}{\partial V} = \frac{\partial \mathcal{L}_1}{\partial S} = \frac{\partial \mathcal{L}_1}{\partial \lambda_1} = \frac{\partial \mathcal{L}_1}{\partial \lambda_2} = 0 \tag{20}$$

must be applied to find the extremized quantities.

## 5. Conclusions

Growing biological systems were modeled in generic terms and investigated using partial derivatives and Lagrange multipliers. Grown stock density and temperature sum were used as independent variables. Estate capitalization increased continuously with grown stock and temperature sum, whereas capital return rate and gross profit rate reached a maximum with respect to the grown stock. As two restrictions were applied simultaneously, the results mostly but not always followed intuition. The derivative of capital return rate with respect to gross profit rate was positive, and negative with respect to capitalization. The derivative of capitalization with respect to capital return rate showed some positive and some negative values, as well as that with respect to gross profit rate. The derivative of gross profit rate was positive with respect to both capitalization and capital return rate. The results indicated a variety of alternative strategies, which may or may not be multiobjective.

**Funding:** This work was supported by Niemi-Säätiö [grant IV].

**Institutional Review Board Statement:** Not applicable.

**Informed Consent Statement:** Not applicable.

**Data Availability Statement:** Not applicable.

**Conflicts of Interest:** The author declares no conflict of interest.

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
