# Peer review of "Lagrangians of Multiannual Growth Systems"

_foundations, doi:10.3390/foundations3010011_

Round 1

Reviewer 1 Report

Paper is short and concise, but includes all that is needed. I have only two remarks, will it be possible to divide current Introduction into proper Introduction about forest industry, forest estates, temperature sums and literature study, but it is really minor issue. A little more important is lack of source for Table 1 on page 4. Those numbers are interesting, but stating that they come from European boreal forestry is a little too vague.

Author Response

Thank you very much!

The author fully agrees that Table 1 required more justification. Now it is explained in the text that the parameters were adjusted to match Eqs. (2) and (8) to boreal growth and yield data [references], as well as to financial data from the region [references].

Yes, the primary application obviously is in forestry, and Table 1 contains parameters adjusted for boreal forestry. However, this author aspires generality in this paper, and would like to avoid making it a specific forestry paper.

It is worth noting that another Reviewer requested quite a few additional specifications. Correspondingly, there are quite many developments in the text, as well as a few Equations in the Discussion. The author hopes this has improved the readability of the paper.

Thank you again for your constructive criticism.

Reviewer 2 Report

I liked the article I reviewed, in the article the author considered a universal model of estates growing perennial plants using hyperbolic growth functions and Lagrange multipliers. I have no questions about the sections of the article indicated by the author, they reveal the topic of the manuscript in a meaningful way. The author considers a simple expression for the capitalization of forest lands, as well as for the rate of gross profit and the rate of return of capital. Further, capitalization is taken as the main objective function, and the rate of return on capital and the rate of gross profit are taken as restrictions. Then the gross profit margin is maximized, while capitalization and the rate of return of capital are restrictions. The results obtained by the author indicate a variety of alternative strategies that may or may not be multiobjective. I recommend this manuscript for publication in the journal.

Author Response

Thank you very much!

It appeared that Table 1 required more justification. Now it is explained in the text that the parameters were adjusted to match Eqs. (2) and (8) to boreal growth and yield data [references], as well as to financial data from the region [references].

It is worth noting that another Reviewer requested quite a few additional specifications. Correspondingly, there are quite many developments in the text, as well as a few Equations in the Discussion. The author hopes this has improved the readability of the paper.

Thank you again for your constructive criticism.

Reviewer 3 Report

The notation and its references trough the manuscript should be homogenous:

 Independent variables:

V = average grown stock (line 64) €/ha; grown stock expected value; it is not treated as a random variable

S = average growing season temperature sum

Dependent variables

P = average profit rate   (line 76) average gross profit rate (line 73), expected value of gross profit rate (figure 1); the last name seems to imply that you treat it as a random variable, but in (2), it is a deterministic function of V; although, later on, when you define the scale factor 'a' as a function of S (7), P will depend on V and S, but again, as a deterministic function. Later in the discussion, the 'capital return rate' is surely the operative return rate R; the reader must guess about.

C = capitalization €/ha    (line 69)  as a linear function of V, but also on S depending of land value assumption: B = bare land value = βS

R = average operative return rate = P/C  (line 79) 

Three 'optimization' problems are presented, but, there no seems to be any optimization to be done, as these are functions with 2 variables and 2 restrictions.

 max R(V, S)  with two restrictions  P = p and C = c

But if R = P/C is a function of V and S and you introduce two restrictions on P(V, S) = p and C(V, S) = c, you get R = p/c as the only value for R, and this value you call it R*? And the values of V and S could be deduced from the two restrictions? C* is then c?

 max C(V, S)  with two restrictions R = r and P = p

Again just with the restrictions you can obtain directly V and S, and C = p/r; surely this value is C*?

 max P(V, S) with the restrictions C = c and R = r

 We obtain directly P = cr; is this P*?

You tell (line 132) that analytical solutions exist; but they are derived directly from the restrictions, in the three cases.

And the restrictions are not presented/justified.

But later on you give values to several parameters; for example, from (2) and table 1, P = 0.1(1 - tanh(0.000101V))V, that is P is depending on V but not on S. So the maximization of P in (14) is not clear; equating (5) to cero would determine V without using any restriction.

Regarding the figures, you present the functions to be maximized dependent on V for three levels of S: the axes should be labelled using the names previously defined for the variables: for different levels of S

Figure 1: P(V)

Figure 2: C(V)

Figures 3, 4: are the derivatives of P and C with respect to S, as stated (lines 161,…) as a function of V? (for different values of S)

Figure 5: in the title it seems that R is plotted depending on P; but then V is in the X-axis; and in the caption, is the derivative of C or P?

 And so on.

This is to be clarified, the figures properly labelled, and the variables referred always with the same names, as well as the non-existence of maximization problems, as presented.

Author Response

Thank you very much. The thoughtful comments of this reviewer provide an opportunity to critically evaluate the content of the analysis, as well as the comprehensibility of the expression.

It is now explained in the text that the independent variables are not random but determined by agent actions. The average grown stock is determined by the amount of harvesting, which decreases the grown stock magnitude. It is also explained that the amount of the grown stock necessarily fluctuates in time, since harvesting occurs discretely, instead of continuously. Such fluctuation however is considered unimportant in the present context, as it evens out along with time. The temperature sum is predominantly determined by estate location, but it may vary with time. Correspondingly, any agent is able to contribute to this variable by estate acquisition.

It is now explained in the text that dependent variables are given in terms of deterministic Equations since only the expected values are here of interest; fluctuations of the dependent variables tend to average out along with time.

Indeed, three optimization problems are presented. Any of them is solvable since there are two independent variables that appear in two Equations. The Lagrangians appearing in Eqs. (12), (13) and (14) are not optimized. Optimization occurs by extremizing the Lagrangians, as explained on lines 127, 133, and 139. It is now explained that all quantities with the asterisk * as a superscript correspond to the extremized quantities. It also is now explained that extremizing with respect to the Lagrange multipliers (line 127) results in the extremized functions getting values that equal the restrictions.

The observation by the Reviewer that R* = P*/C* = p/c is very significant, and it does require clarification in the text. The relation of this fact to the reported results is now explained at the end of the Discussion.

The Reviewer appears to have missed the relation between Eq. (2) and Table 1. The Quantity given in Table 1 is “alpha”. The Quantity appearing in Eq. (2) is “a”. Then, Equating Eq. (5) to zero, the result depends on the temperature sum. But yes, without any restriction, the profit rate can be maximized for any given temperature sum. The Lagrangians are then needed for extremizing quantities in the presence of restrictions.

Thank you for hinting about the Figure labels. They have now been edited.

Round 2

Reviewer 3 Report

You present three optimization problems with restrictions:

The first about the operative return rate R, that depends on the grown stock V and the sun hours S.

 max R  =P/C = 0.1V tanh ((S - 600)/450) (1 - tanh (0.000101V)) / [0.7 S +  (1 + 0.3)V]

with two restrictions, on P = p and C = c. Are the values p and c fixed, or they are variables introduced in the model? In the graphs, they are used as parameters than can vary.

These restrictions put a limit on the return rate and on the capitalization. I can not find where these limits have been determined, or the reasons that the profit or the capitalization should be assigned the values p and c.

 But, as P =  0.1V tanh ((S - 600)/450) (1 - tanh (0.000101V)) and C = 0.7 S +  (1 + 0.3)V, if you fix the two values p and c, from these equations, you can obtain S = s and V = v directly, so, I don't see where the optimization problem is. You will get R as a single point depending on p and c.

 When you formulate the Lagrangian function, L1(R, λ1, λ2) and maximize it, the restrictions are included as equalities, P = p and C = c. and this is where just solving both equations, you should get the corresponding values V = v and S = s, and thus r = R(v, s) = R(p, c).   

 The same can be said about other maximization processes.

If you introduce fixed restrictions, the problem is trivial; if you consider the restriction limits as variables, the different functions will attain the optimum as a function of the restriction values. I think this should be stated more clearly, but then no need of any Lagrangian, derivatives and so on.

 In figure 1, you represent the funcion profit rate, P, for 3 values of S. and, as x-axis you use the capitalization restriction of the value/ha, c. So, as c increases, P begins to grow, and then, decreases. But this happens because P (depending of a sigmoidal type function, as the tanh) is a function of V, and there is the relation c = (0.7S) + 1.3V (when I try to reproduce this función, the máximun is shifted to the right). But, if you used fixed values for S, the functions to optimeze are just function of V.

Author Response

The restrictions r, p and c , introduced in Eqs. (12), (13) and (14), may vary. There are no limits for these restrictions, neither do they have any fixed values. That is why they are not given anywhere. This is a standard appearance of the Lagrangians.

There are no restrictions “s” or “v”. Correspondingly, no Equations such as “V = v” or “S = s” exist.

The Lagrangian treatment does not intend clarify the values of R, P or C.
It clarifies the values of the total derivatives dR/dp, dR/dc, dP/dr, dP/dc, dC/dr, and dC/dp.
The values of these total derivatives are shown in Figs. 5 to 10. The total derivatives are expressing the change of any objective function under the change of any one of the applied two simultaneous restrictions.
Needless to say, the sensitivity of the objective functions to the restrictions depends on the model parameters Grown Stock (V) and Temperature Sum (S), as is shown in Figs. 5 to 10.

The text at the end of the Introduction is now modified, to clarify the above.

Round 3

Reviewer 3 Report

It is not clear the optimization process. The independent variables are V and S. and you define the 3 objective functions

C(V, S) = βS + (1 + u)V

P(V, S) = α tanh((S – ζ)/φ) V (1 – tanh(kV))

R(V, S) = P/C

For example, when maximizing R, two restrictions are formulated: P = p and C = c.  Using the definitions of C and P, it is equivalent of  formulating restrictions on S = s and V = v. But imposing these two conditions, that would hold as identities (even if you maximize the first Lagragian function), so, R = p/c = r, so, there is no optimization, but one unique value R = r, that depends on p and c.

Again: if you force these restrictions on the definitions of

c = C(V, S) = βS + (1 + u)V

p = P(V, S) = α tanh((S – ζ)/φ) V (1 – tanh(kV))

V and S  will be determined, such as c = C(v, s) and p = P(v, s).

Considering the function R, as a known function of P and C, and, thus, on V and S, to, there is no problem of obtaining directly the derivatives. And these would depend on v and s.